# Yucatan Miniswine Model of Atrial Fibrillation: Clinical Relevance

**Pouria Mostafizi**[1], **Eli Lefkowitz**[1,2], **Jacob Ref**[1,2], **Fox Bravo**[1,2], **Daniel Benson**[1,2,3], **Deirdre O'Donnell**[1,2], **Kenneth Fox**[1,2,4], **Adrian Grijalva**[1], **Mary Kaye Pierce**[1], **Grace Gorman**[1], **Alice McArthur**[1], **Sherry Daugherty**[1], **Anil Shendge**[5,6], **Amitabh C. Pandey**[5,6], **Jordan Lancaster**[1], **Jen Watson Koevary**[1], **Steven Goldman**[1]*, **Talal Moukabary**[1]

1 Sarver Heart Center, University of Arizona, Tucson, Arizona, United States of America, 2 College of Medicine, University of Arizona, Tucson, Arizona, United States of America, 3 Dept Biomedical Engineering, University of Arizona, Tucson, Arizona, United States of America, 4 College of Medicine, Department Surgery, University of Arizona, Tucson, Arizona, United States of America, 5 Section of Cardiology, Department of Medicine, Southeast Louisiana VA Medical Center, New Orleans, Louisiana, United States of America, 6 Department of Medicine, Tulane University School of Medicine, New Orleans, Louisiana, United States of America

* goldmans@shc.arizona.edu

## Abstract

### Introduction

Persistent atrial fibrillation (AFib) is the most common chronic arrhythmia in adults in the United States and is associated with significant morbidity, including thromboembolic events, stroke, and heart failure. Despite available therapies such as catheter ablation and antiarrhythmic drugs, AFib remains incurable for many patients. Our study aims to develop a large-animal model of AFib in Yucatan miniswine to support investigation of new therapeutic approaches for this disease.

### Methods

Yucatan miniswine were selected for their physiological similarity to humans and suitable size for long-term studies. Each animal was initially fitted with an external FitBark 2.0 collar to track activity as a surrogate for quality of life. Animals then underwent implantation of an atrial pacing lead in the right atrial appendage, a pacing generator, and an insertable cardiac monitor (ICM Reveal LINQ™) implanted subcutaneously along the left scapula. One week after the pacemaker implantation, animals underwent rapid atrial pacing to induce persistent AFib. All procedures were performed in accordance with relevant institutional and regulatory guidelines.

### Results

Atrial fibrillation was successfully induced in 4 of 6 animals within 80.3±22.3 days of initiating pacing with three animals going into persistent AFib and one animal going

**Data availability statement:** All relevant data are within the paper and its Supporting Information files. Data repository can be found at 10.25422/azu.data.30471077.

**Funding:** This work was supported by Tech Launch Arizona Asset Development Grant UA17-095, the WARMER Research Foundation, and the Sarver Heart Center, University of Arizona." "The funders had no role in study design, data collection and analysis, decision to publish, or preparation of the manuscript.

**Competing interests:** The authors have declared that no competing interests exist.

into paroxysmal AFib. The definition of persistent AFib was that animals remained in AFib for more than 14 days after pacing was discontinued. Paroxysmal AFib was defined as AFib lasting less than 14 days. Activity levels decreased following persistent AFib onset, indicating a decline in overall health and quality of life. Histopathological analyses showed significant increases in fibrosis and loss of atrial cardiomyocytes after persistent AFib was induced in swine. Several anatomical and technical challenges, particularly related to vascular access and cardiac dimensions, were overcome through customized surgical strategies, including jugular venous cutdowns, lateral cervical ICM implantation, long vascular sheaths, custom styluses, and perioperative antibiotic coverage. These innovations were critical to establishing a robust and reproducible persistent AFib model.

## Introduction

Atrial fibrillation (AFib) is the most common sustained cardiac arrhythmia in adults, currently affecting an estimated 2.7 million people in the United States and increasing in prevalence as populations age and cardiovascular survival improves [1]. AFib is strongly associated with stroke, heart failure, diminished functional capacity, and increased mortality, and it often exacerbates left ventricular dysfunction [2]. Once established, AFib becomes self-perpetuating with atrial stretch, inflammation, heart failure, and progressive fibrosis driving both electrical and structural remodeling, creating a substrate that promotes this persistent arrhythmia [3]. Although advances such as pulsed-field ablation and antiarrhythmic agents have improved outcomes for select patients, AFib incidence and recurrence remain high, reflecting a critical need for therapies that directly address the underlying atrial pathology, specifically fibrotic replacement of functional atrial cardiomyocytes.

Large-animal models of cardiac disease are essential for bridging mechanistic insights to clinical intervention, yet current preclinical AFib models rarely reproduce the chronicity, fibrosis, and electrophysiologic complexity observed in human disease. These limitations motivated our development of a robust, reproducible Yucatan miniswine model of AFib that more faithfully captures the human condition. Yucatan miniswine have cardiac anatomy, chamber size, conduction system organization, and fibrotic remodeling patterns that closely mirror those of humans, making them ideal for translational arrhythmia research.

Here, we describe a novel miniswine model of AFib designed to enable longitudinal monitoring, mechanistic study, and therapeutic testing. A unique aspect of our work is the inclusion of continuous electrocardiographic recordings obtained via implanted internal cardiac monitors (ICMs) to confirm arrhythmia induction, track AFib burden, and document persistence over time. In parallel, FitBark activity trackers were used to quantify behavioral and functional changes in the animals between sinus rhythm and AFib, adding clinically relevant physiologic endpoints. Together, these tools create a comprehensive platform that recapitulates key clinical features

of human AFib and provides a rigorous foundation for evaluating new interventions that potentially target atrial fibrosis the underlying cause of the disease.

## Materials and methods

### Animal studies, ethical considerations

All animal work was conducted under the oversight of the University of Arizona Institutional Animal Care and Use Committee (IACUC) and the University of Arizona Animal Care (UAC) veterinary staff. The IACUC oversees the university's animal care and use program and is responsible for reviewing and approving all activities involving vertebrate animals for research, teaching, and testing. Compliance information includes USDA, Class R Research Facility, Registration Number: 86-R-0003, NIH/OLAW Assurance Number: D16-00159, and AAALAC International Accreditation since 1969, Accreditation Number: 000163, Status: Continued Full Accreditation. Animals were observed at least once daily, with increased monitoring following surgical procedures. The protocol described in this peer-reviewed article is published on protocols. io, DOI: x.doi.org/10.17504/protocols.io.yxmvm1p76v3p/v1 and is included for printing as supporting information file 1 with this article.

### Animal preparation

Approximately one-year-old Yucatan miniswine (N=6) were enrolled in this study. All procedures were approved by the Institutional Animal Care and Use Committee (IACUC) and adhered to ARRIVE guidelines. Animals were housed in a temperature-controlled facility with a 12-hour light-dark cycle. Standard diets and ad libitum water access were provided in the enriched environment to promote well-being. The study adhered to USDA and AWA regulations, and all efforts were made to minimize animal distress.

### Anesthesia and surgical approach

Anesthesia was induced using ketamine (11–33 mg/kg IM) and/or midazolam (0.1–0.5 mg/kg IM). Isoflurane (1−3%) was administered for maintenance anesthesia for the duration of the procedure. Animals were administered buprenorphine Sr 0.12–0.27 mg/kg for minimizing pain and distress after surgical implantation of the pacing device. Standard aseptic techniques were followed during surgical preparation. The ICM was implanted subcutaneously in the left chest. For pacemaker implantation, the right external jugular vein was accessed using a cutdown with preemptive loose ligation to minimize bleeding. Custom elongated sheaths and stylus shapes were used to position the pacing atrial lead in the right atrial appendage. A Medtronic CRT-P device was implanted subcutaneously in a small pocket in the right chest. Proper lead placement was confirmed with a multi-lead ECG.

### Pacing protocol: Pacemaker activation

The pacemaker was activated one-week post-implantation. Rapid atrial pacing was initiated at 400 bpm (150 ms cycle length). If AFib was not sustained, the pacing rate was gradually increased to 600 bpm (100-ms cycle length) in 10-ms increments. The ICM Device, the pacemaker generator, the programmer (Medtronic 2090), and Medtronic CareLink Encore are shown in Fig 1. An illustration of how all these devices work together in the swine model is shown in Fig 2. Activity levels were tracked 24/7 with the FitBark activity collars show in Fig 3.

### Euthanasia

Animals were euthanized at endpoints via potassium chloride injection under deep anesthesia. Authorization was confirmed with no heartbeat on the ECG, prior to tissue harvest/necropsy.

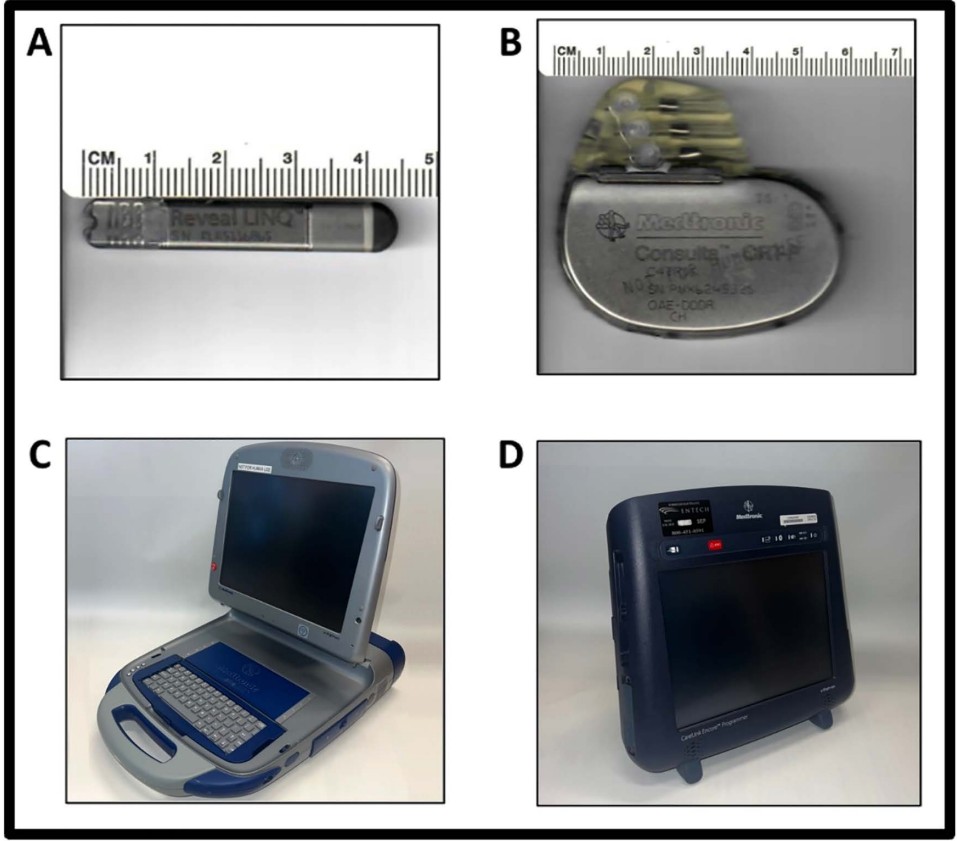

**Fig 1. Medical Devices. (A)** Internal Cardiac Monitor Device. **(B)** Pacemaker Device. **(C)** Programmer (Medtronic 2090) sets regular, rapid atrial overdrive pacing parameters on the pacemaker (400-600 bpm, 8V). **(D)** The programmer (Medtronic CareLink Encore) is used to record frozen strips and episodes from the Internal Cardiac Monitor, which constitutes Internal Cardiac Monitor reports.

## Histology

Cardiac biopsies were taken from the left and right atrium, fixed in 10% formalin and paraffin embedded before undergoing histopathological evaluation. Specimens were fixed in 10% neutral buffered formalin, transferred to 70% ethanol, and processed for paraffin embedding and sectioning. Histologic sections were stained with Masson's trichrome to assess myocardial architecture and fibrosis.

## Imaging

Images of stained tissue sections were captured using a Leica DMI6000B motorized inverted microscope under brightfield illumination and a 10x objective. All imaging parameters, including exposure time, white balance, and focus, were standardized across samples to ensure consistency in image quality. Tile scans were stitched together using Leica Application Suite X (LAS X) software. Images were acquired through the University of Arizona Optical Imaging Core Facility (RRID:SCR_023355).

## Image quantification

Images were analyzed in ImageJ to measure collagen area. The scale was set with distance = 0, known distance = 0, and pixel aspect ratio = 1.0. Images were converted to an RGB stack, and thresholds were adjusted in the blue channel to

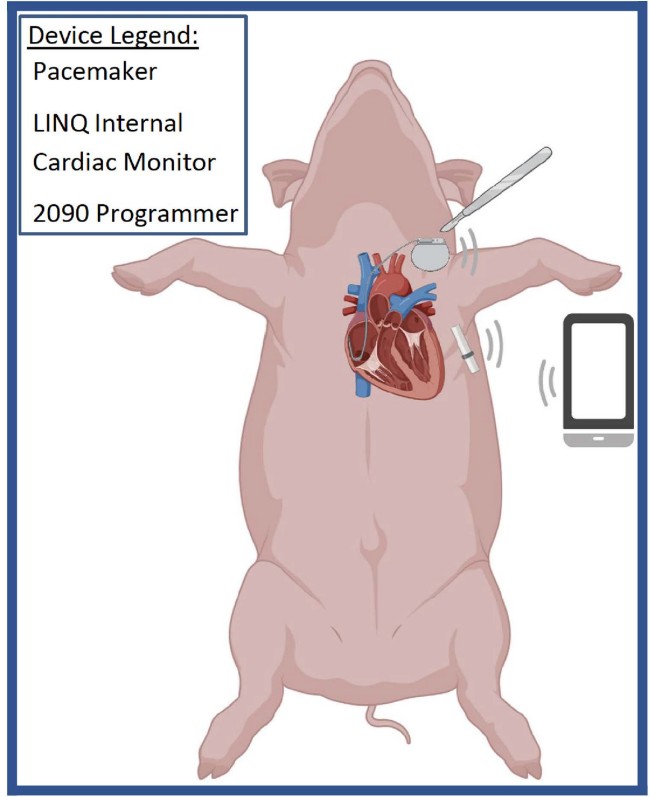

**Fig 2. Medtronic Rapid Atrial Pacing System.** A pacing wire is inserted into the right atrial appendage and connected to a pacing generator implanted in the lateral neck wall muscle. A programmer, Medtronic CareLink 2090, controlled the pacing generator and internal cardiac monitor wirelessly. https:// biorender.com/mngrgm4.

isolate blue and red pixels. Thresholds for blue pixels were set to 130–210 for control tissue and 180–210 for AFib tissue, while thresholds for red pixels were set to 0–100 for control tissue and 0–150, respectively. Collagen and Muscle area were quantified relative to total tissue area. Statistical analysis was performed in SigmaPlot using an unpaired t-test with a significance of $P < 0.05$.

## Results

Several procedural challenges required iterative refinement to ensure reliable pacing and consistent arrhythmia induction. Surgical adaptations, including loose ligation to reduce bleeding, use of custom elongated sheaths, and a pull-back lead-delivery technique improved pacing stability, while multi-lead ECG confirmation ensured accurate lead positioning.

### Outcomes assessment

Atrial fibrillation was successfully induced in 4 of 6 (67%) animals within 80.3±22.3 days of initiating pacing with 3 animals going into persistent AFib and one animal going into paroxysmal Afib as demonstrated by electrocardiogram (Fig 4) and Kaplan-Meier curve (Fig 5). Complication rates, including spontaneous conversion to sinus rhythm and rapid ventricular response, were documented and addressed through protocol modifications. One animal spontaneously reverted to sinus rhythm, and one died from ventricular tachycardia related complications. Ventricular response rates during pacing varied widely among animals (100–180 bpm); one animal had dramatically elevated ventricular responses close to 200 bpm

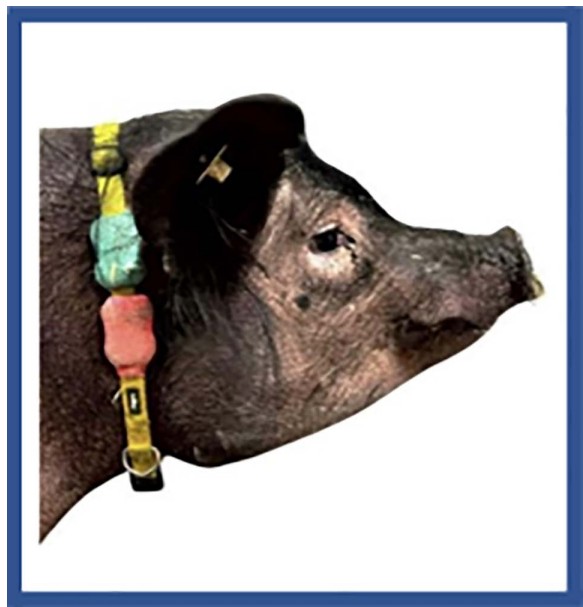

**Fig 3. Activity Level Monitoring.** Activity levels 24/7 were tracked using Fitbark collars worn by the swine as illustrated above.

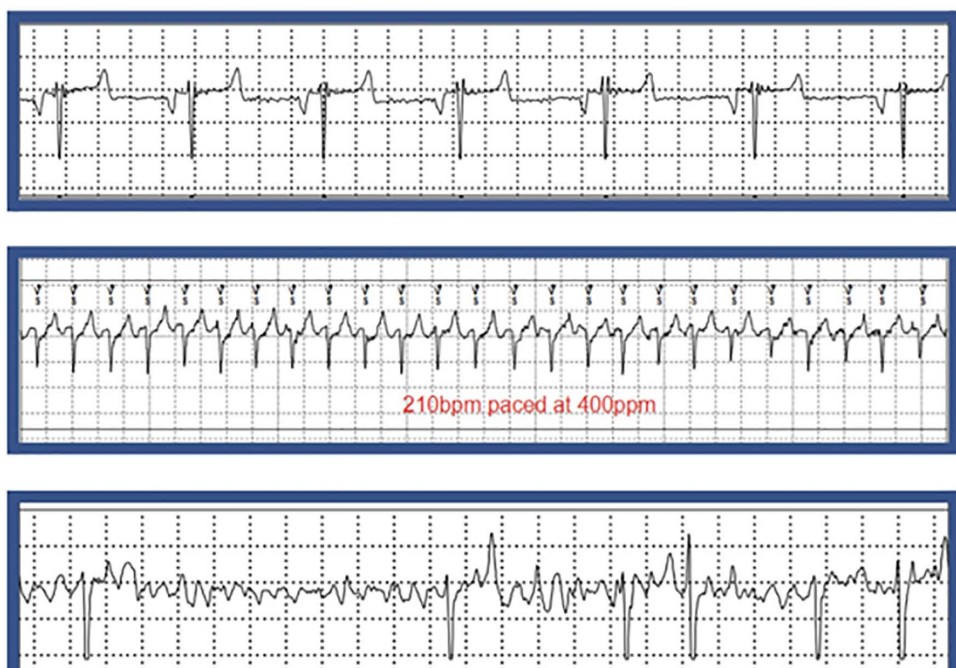

**Fig 4. Cardiac Monitoring by Electrocardiograms.** Electrocardiogram strip extracted from implanted cardiac monitor displaying. Normal Sinus Rhythm, Paced Rhythm, and Atrial Fibrillation (AFib).

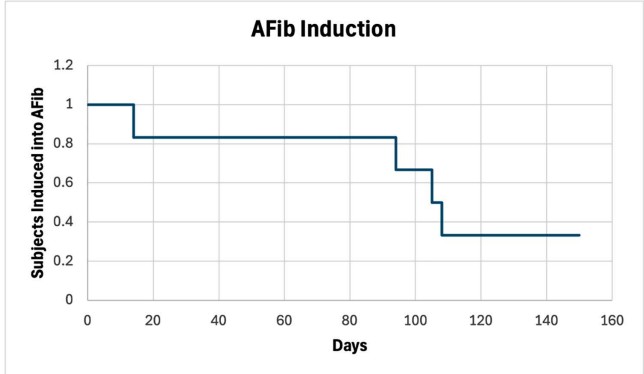

**Fig 5. AFib Induction Time.** Kaplan-Meier curve depicting the proportion of pigs remaining free from induced AFib over this observation period (n = 6). Four of six pigs (67%) successfully converted to AFib during the study period, with conversions occurring at 14-, 94-, 105-, and 108-days post-induction attempt. Two pigs (33%) did not develop AFib during the observation window. The stepped curve represents individual AFib induction events, with the y-axis showing the proportion of subjects not yet induced into AFib and the x-axis showing time in days.

and was refractory to beta-adrenergic and calcium-channel blockade. Interestingly, the sinus rates during their normal activity varied from sinus bradycardia 40–50 bpm to sinus tachycardia to over 200 bpm. No thromboembolic events were observed even though the animals were not anti-coagulated.

## Protocol adjustments

Protocol refinements minimized mortality and pacing-related complications over time. This included frequent monitoring of each animal's rhythm during pacing. We did not initiate pacing until one week after pacer implantation to ensure that the atrial lead was securely fixed in the right atrial appendage wall. Once we initiated pacing, we would closely monitor each animal's behavior to make sure that they were not suffering any side effects from the pacing. Occasionally pacing initiated hiccups because of intermittent diaphragmatic pacing, that was solved by decreasing the voltage on the unit. One animal went into ventricular tachycardia every time we initiated pacing. Upon necropsy, it was found that the atrial lead was implanted too close to the tricuspid valve and thus pacing the right ventricle as opposed to the right atrial appendage. The Yucatan model supported multi-week maintenance of persistent AFib, contrasting with traditional domestic pig models that often-required euthanasia after approximately 20 days due to heart-failure [4]. Comparable in chronicity to more recent Landrace tachypacing studies achieving ~6 weeks of AFib, this model offers the advantages of smaller, more robust animals and continuous rhythm monitoring, enabling extended-duration studies of persistent AFib [5].

## Activity monitoring

Activity levels documented 24/7 with the FitBark activity trackers showed decreased activity with the occurrence of persistent AFib (Fig 6). To our knowledge this is the first report of documented decreases in activity in an animal model of AFib. This shows the clinical relevance of this model, i.e., being able to equate the presence of persistent AFib with a decrease in quality of life in an animal model.

## Histological evaluation

Masson's trichrome staining revealed myocardial architecture differences in animals that were induced into persistent AFib. Overall, blue staining on Masson's trichrome reflects collagen-rich extracellular matrix and fibrous connective tissue, including interstitial and perivascular extracellular matrix (ECM) components. Red staining on

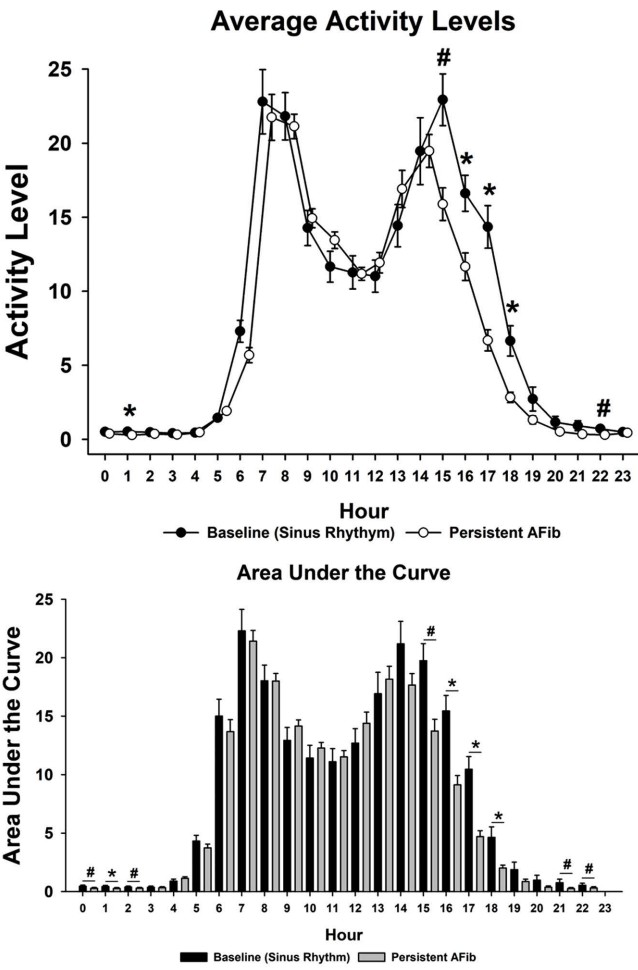

**Fig 6. AFib Activity Level.** Hourly activity levels, tracked using Fitbark collars show a significant decrease during AFib episodes compared to baseline, mirroring human symptoms like fatigue and exercise intolerance. #P < 0.01, *P < 0.05 refers to persistent AFib versus baseline.

Masson's trichrome highlights muscle fibers and cytoplasmic proteins, predominantly reflecting viable myocardium, including cardiomyocytes and smooth muscle cells. Our data display an increased fibrosis within the ECM and a significant reduction in cardiac muscle tissue (Fig 7). This pattern reflects loss of functional atrial muscle and expansion of collagen-rich ECM, a substrate that impairs atrial contractility, disrupts electrical conduction, and promotes AFib maintenance, thereby supporting the clinical relevance and translational fidelity of this large-animal model.

## Discussion

This study introduces a technology-enhanced Yucatan miniswine model of persistent AFib that incorporates continuous rhythm and activity monitoring. The use of Medtronic Reveal LINQ™ implantable internal cardiac monitors provided uninterrupted ECG data, enabling precise characterization of AFib onset, duration, and ventricular response in freely moving animals. This approach overcomes limitations of prior large-animal AFib models, which often relied on intermittent surface ECGs or short-duration telemetry and consequently captured only limited snapshots of arrhythmia burden.

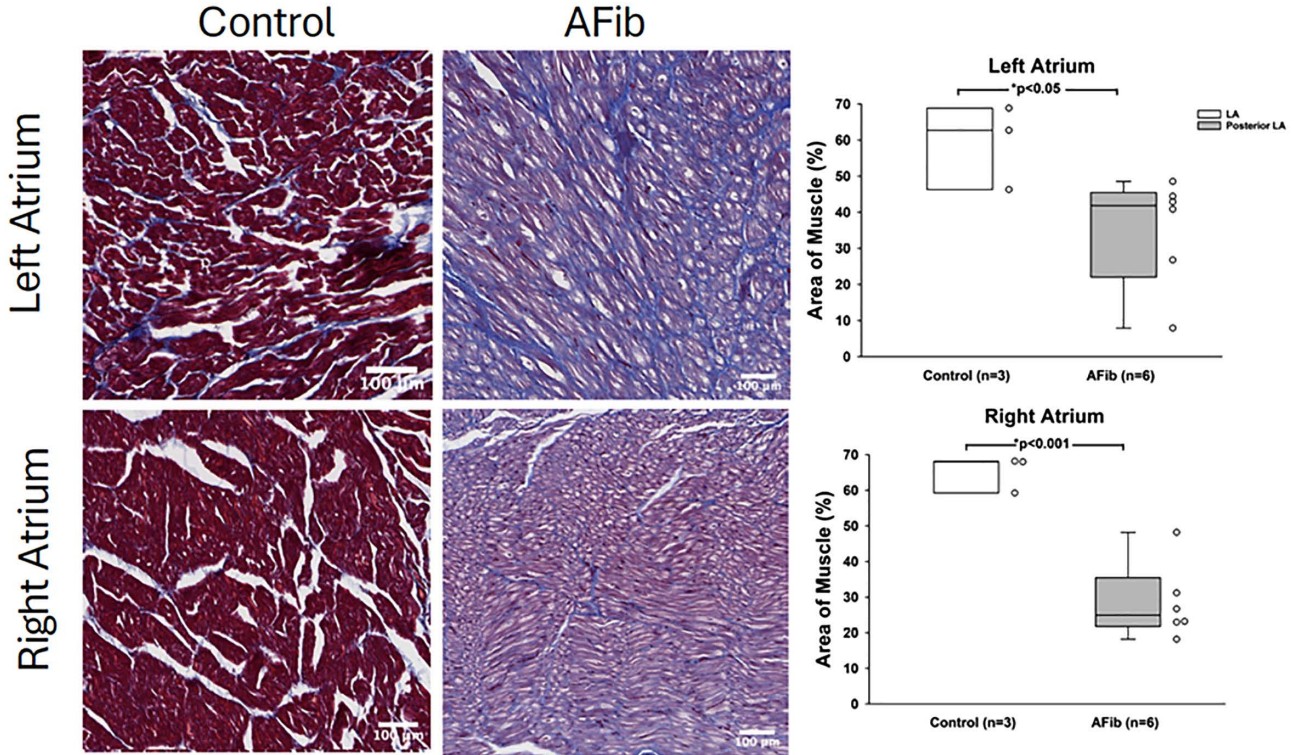

**Fig 7. Histopathological Analysis of persistent AFib tissue. Masson's trichrome staining of left and right atrial biopsies were imaged and analyzed for blue/red positive staining in ImageJ.** Swine that were induced into persistent atrial fibrillation displayed increased collagen staining and significantly reduced muscle staining ($P < 0.05$) and ($P < 0.001$).

The addition of Fitbark 2 accelerometer collars supplied continuous, quantitative behavioral data without the need for handling or restraint. Previous AFib models have generally lacked objective assessments of spontaneous physical activity or functional changes during arrhythmia. In contrast, the combined monitoring approach used here produced a comprehensive dataset linking presence of persistent AFib with real-time activity patterns. Notably, animals demonstrated reduced activity during persistent AFib compared with their sinus-rhythm baseline, providing an objective indicator of functional change associated with the arrhythmia (Fig 6). This observation is consistent with well-established associations between AFib and reduced exercise capacity in clinical populations [6–7] and supports the relevance of activity measurements as a surrogate for quality of life and functional decline in preclinical studies.

The ability to track both electrical and behavioral parameters enhances the utility of this model for evaluating therapeutic interventions. Continuous activity monitoring offers a functional endpoint that complements rhythm analysis and permits assessment of whether an intervention affects both AFib burden and activity levels. This dual-parameter approach aligns with contemporary clinical evaluation of AFib therapies, which need to incorporate functional and quality of life metrics alongside rhythm outcomes.

## Histological analyses

Persistent AFib in our model was associated with clear structural remodeling of the atrial myocardium. Quantitative analysis of Masson's trichrome staining of left and right atrial biopsies demonstrated an increase in fibrosis with a corresponding significant reduction in atrial myocardial tissue, indicating replacement of functional myocardium by fibrotic scar

(Fig 7). This pattern of enhanced fibrosis and myocyte loss is consistent with atrial cardiomyopathy observed in patients with longstanding AFib and supports that our swine model recapitulates the histopathological substrate that sustains and perpetuates persistent atrial fibrillation with high fidelity.

## Comparison with prior AFib models

Earlier AFib models utilized domestic farm pigs, goats, and dogs [8]. Domestic swine breeds like Yorkshire or Landrace grow fast, complicating long-term studies. In contrast, Yucatan miniswine remains smaller, making extended follow-up and housing easier. Goats (~50–70 kg) were the first species used for chronic AFib, while canines, such as beagles, have been used for pacing-induced AFib. The Yucatan model balances human-like cardiac anatomy and a smaller, more manageable size, improving long-term study practicality and animal welfare, challenges domestic pig models struggled to overcome.

There is recent work on including transgenic or optogenetic AFib models. Importantly, this work has been done mice, rats, and in isolated atrial tissue, where investigators have developed new ways to study the mechanisms of AFib and potentially can be applied to study control electrical activity the role of rotors [9–10]. These approaches will potentially help develop new therapies based on these platforms, but they ultimately will have to be investigated in large animal models of persistent AFib. Our model provides an ideal prototype where investigators can define the precise onset of AFib, how long it lasts and its effects on clinical activity. Lastly, tissue from our model will provide gene and protein analyses in a clinically relevant model.

Our model induces AFib by chronic rapid atrial pacing of the right atrial appendage at ~400 beats per minute (bpm). This continuous atrial tachypacing approach is modeled after classic experiments in goats, dogs, and pigs, which demonstrate maintaining a high atrial rate for days to weeks can produce self-sustaining AFib through electrical remodeling. In a seminal goat study, AFib became sustained (> 24 hours) after about 7 days of continuous rapid atrial pacing in 10 of 11 goats [3]. Similarly, chronic right atrial pacing at 400 bpm for 6 weeks in a canine model created a substrate where AFib was easily inducible and maintained for a long duration [11].

Previous swine models used different pacing strategies with varying success. Early attempts at AFib in pigs often involved burst pacing, delivering short, high-frequency pulses in the right atrium to trigger fibrillation. While such bursts could initiate AFib episodes, they usually lasted only seconds to minutes before self-terminating. With sequential pacing and vagal stimulation (neostigmine) in an acute pig model could result in AFib, episodes lasted only ~3.6 minutes on average, with 78% spontaneously reverting to sinus rhythm [12]. Without prolonged pacing, standard domestic pigs struggled to maintain persistent AFib. A 2004 study achieved sustained AFib in pigs using extreme atrial burst pacing at 42 Hz (~2500 bpm) with an implanted pacemaker [4]. AFib became persistent after ~5 days but resulted in a rapid ventricular response (~274 bpm), leading to tachycardia-induced cardiomyopathy and heart failure within 2–3 weeks. Our previous work with other cardiac diseases in Yucatan miniswine shows how valuable this Yucatan model is for studying heart failure after myocardial infraction, developing immune modulation to treat heart failure and effects of lack of socialization on physical activity during COVID [13–16]. Additionally, our team has recently completed a comprehensive review of AFib animal models across both large and small animal species [17].

## Limitations

Additional histological, echocardiographic, and magnetic resonance imaging (MRI) analyses could be investigated to further characterize atrial remodeling and fibrosis associated with this model. Importantly the pacing device and custom software used in this study are no longer available, which may affect direct replication of the stimulation protocol. Despite these constraints, the methods described here provide a robust and reproducible platform for studying AFib pathophysiology and assessing emerging therapeutic strategies in a large-animal model.

## Author contributions

**Conceptualization:** Pouria Mostafizi, Eli Lefkowitz, Jacob Ref, Adrian Grijalva, Amitabh C. Pandey, Steven Goldman, Talal Moukabary.

**Data curation:** Pouria Mostafizi, Eli Lefkowitz, Daniel Benson, Deirdre O'Donnell, Kenneth Fox, Adrian Grijalva, Sherry Daugherty, Anil Shendge, Amitabh C. Pandey, Steven Goldman, Talal Moukabary.

**Formal analysis:** Pouria Mostafizi, Eli Lefkowitz, Jacob Ref, Deirdre O'Donnell, Adrian Grijalva, Sherry Daugherty, Amitabh C. Pandey, Jen Watson Koevary, Steven Goldman, Talal Moukabary.

**Funding acquisition:** Jordan Lancaster, Steven Goldman, Talal Moukabary.

**Investigation:** Pouria Mostafizi, Eli Lefkowitz, Kenneth Fox, Adrian Grijalva, Mary Kaye Pierce, Sherry Daugherty, Jordan Lancaster, Jen Watson Koevary, Steven Goldman, Talal Moukabary.

**Methodology:** Pouria Mostafizi, Eli Lefkowitz, Jacob Ref, Fox Bravo, Sherry Daugherty, Steven Goldman, Talal Moukabary.

**Project administration:** Pouria Mostafizi, Kenneth Fox, Adrian Grijalva, Alice McArthur, Sherry Daugherty, Jordan Lancaster, Jen Watson Koevary, Steven Goldman, Talal Moukabary.

**Resources:** Pouria Mostafizi, Fox Bravo, Adrian Grijalva, Sherry Daugherty, Jordan Lancaster, Steven Goldman, Talal Moukabary.

**Software:** Jordan Lancaster, Jen Watson Koevary, Steven Goldman, Talal Moukabary.

**Supervision:** Kenneth Fox, Adrian Grijalva, Mary Kaye Pierce, Alice McArthur, Sherry Daugherty, Amitabh C. Pandey, Jordan Lancaster, Jen Watson Koevary, Steven Goldman, Talal Moukabary.

**Validation:** Pouria Mostafizi, Eli Lefkowitz, Fox Bravo, Daniel Benson, Adrian Grijalva, Mary Kaye Pierce, Sherry Daugherty, Anil Shendge, Steven Goldman, Talal Moukabary.

**Visualization:** Pouria Mostafizi, Eli Lefkowitz, Fox Bravo, Adrian Grijalva, Sherry Daugherty, Steven Goldman, Talal Moukabary.

**Writing – original draft:** Pouria Mostafizi, Eli Lefkowitz, Jacob Ref, Daniel Benson, Sherry Daugherty, Steven Goldman, Talal Moukabary.

**Writing – review & editing:** Pouria Mostafizi, Jacob Ref, Fox Bravo, Daniel Benson, Deirdre O'Donnell, Grace Gorman, Steven Goldman, Talal Moukabary.

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
