## [Decision Letter · Decision Letter 0]

11 Jul 2025

Yucatan Mini Swine Model of Persistent Atrial Fibrillation: Clinical Relevance

PLOS ONE

Dear Dr. Mostafizi,

Thank you for submitting your manuscript to PLOS ONE. After careful consideration, we feel that it has merit but does not fully meet PLOS ONE’s publication criteria as it currently stands. Therefore, we invite you to submit a revised version of the manuscript that addresses the points raised during the review process.

We look forward to receiving your revised manuscript.

Kind regards,

Kamal Sharma

Academic Editor

PLOS ONE

Journal Requirements:

2. To comply with PLOS One submissions requirements, in your Methods section, please provide additional information regarding the experiments involving animals and ensure you have included details on (1) methods of sacrifice, (2) methods of anesthesia and/or analgesia, and (3) efforts to alleviate suffering.

“This work was supported by Tech Launch Arizona Asset Development Grant UA17-095, the WARMER Research Foundation, and the Sarver Heart Center, University of Arizona.”

5. We note that your Data Availability Statement is currently as follows: All relevant data are within the manuscript and in Supporting Information files.

8. We note that Figure 1 in your submission contain copyrighted images. All PLOS content is published under the Creative Commons Attribution License (CC BY 4.0), which means that the manuscript, images, and Supporting Information files will be freely available online, and any third party is permitted to access, download, copy, distribute, and use these materials in any way, even commercially, with proper attribution. For more information, see our copyright guidelines: http://journals.plos.org/plosone/s/licenses-and-copyright.

 1. You may seek permission from the original copyright holder of Figure1  to publish the content specifically under the CC BY 4.0 license.

9. Please ensure that you refer to Figures 1-4 in your text as, if accepted, production will need this reference to link the reader to the figure.

10. Please include captions for your Supporting Information files at the end of your manuscript, and update any in-text citations to match accordingly. Please see our Supporting Information guidelines for more information: http://journals.plos.org/plosone/s/supporting-information .

11. We note you have not yet provided a protocols.io PDF version of your protocol and/or a protocols.io DOI. When you submit your revision, please provide a PDF version of your protocol as generated by protocols.io (the file will have the protocols.io logo in the upper right corner of the first page) as a Supporting Information file. The filename should be S1_file.pdf, and you should enter “S1 File” into the Description field. Any additional protocols should be numbered S2, S3, and so on. Please also follow the instructions for Supporting Information captions [https://journals.plos.org/plosone/s/supporting-information#loc-captions]. The title in the caption should read: “Step-by-step protocol, also available on protocols.io.”

Please assign your protocol a protocols.io DOI, if you have not already done so, and include the following line in the Materials and Methods section of your manuscript: “The protocol described in this peer-reviewed article is published on protocols.io (https://dx.doi.org/10.17504/protocols.io.[...]) and is included for printing purposes as S1 File.” You should also supply the DOI in the Protocols.io DOI field of the submission form when you submit your revision.

If you have not yet uploaded your protocol to protocols.io, you are invited to use the platform’s protocol entry service [https://www.protocols.io/we-enter-protocols] for doing so, at no charge. Through this service, the team at protocols.io will enter your protocol for you and format it in a way that takes advantage of the platform’s features. When submitting your protocol to the protocol entry service please include the customer code PLOS2022 in the Note field and indicate that your protocol is associated with a PLOS ONE Lab Protocol Submission. You should also include the title and manuscript number of your PLOS ONE submission.

Reviewers' comments:

Reviewer's Responses to Questions

**Comments to the Author**



Reviewer #1: Yes

Reviewer #2: Yes

2. Has the protocol been described in sufficient detail?

To answer this question, please click the link to protocols.io in the Materials and Methods section of the manuscript (if a link has been provided) or consult the step-by-step protocol in the Supporting Information files.

Reviewer #1: Partly

Reviewer #2: Yes

3. Does the protocol describe a validated method?

Reviewer #1: Yes

Reviewer #2: Yes

4. If the manuscript contains new data, have the authors made this data fully available?

Reviewer #1: Yes

Reviewer #2: Yes

**5. Is the article presented in an intelligible fashion and written in standard English?**

Reviewer #1: Yes

Reviewer #2: Yes

Reviewer #1: Comments to the authors

This manuscript presents a new Yucatan mini swine model for sustained AFib which overcomes many of the limitations of previous large-animal models by capitalizing on the anatomical and physiological consistency of the Yucatan mini swine with that of humans. The major novelty of the study is that it uses implantable cardiac monitors (Medtronic Reveal LINQ™) and Fitbark 2.0 activity trackers to allow continuous monitoring of both arrhythmia and behavioral changes – a substantial translational value.

The Yucatan swine animal model is well-given, providing several advantages over other domestic breeds in regard to size consistency and potential long-term investigations. Furthermore, the fact that individual surgical approaches and device modifications, made more extensive operations or those with adverse anatomical conditions feasible, is indicative of good technical performance.

Together, the model provides a reliable and clinically applicable system for chronic AFib investigation and therapy assessment, potentially allowing us to better understand the arrhythmia-associated functional impact.

Major Weaknesses

1.The authors should report exact quantitative details and statistical analysis for their main results?

This manuscript does not yet adequately report statistics on the results (time to AFib induction, variability in ventricular rates, extent of activity reduction) and histological changes observed in trial animals (e.g. fibrosis).

I also invite them to add a more complete statistical analysis and the corresponding graphical summaries (e.g., time to AFib induction as well as activity and rate variability, in the form of Kaplan-Meier curves and bar plots with error bars, respectively) to ensure the clarity and strength of your findings.

2.I'd suggest: They could validate their model more thoroughly on the molecular and histopathological level:

The present manuscript yields little evidence for atrial remodeling, fibrosis or gene expression changes which are necessary to support the translational relevance of model to human AFib.

Integration of echocardiographic and histological analysis, and molecular markers of structural and electrophysiological remodeling, will reveal the pathophysiological relationships between the model and clinical AFib.

3.Proper discussion of limitations of the model in the discussion section?

The manuscript is too focused on strengths of the model without enough discussion of limitations such as: variable ventricular rates, n=6, early death of one animal, and its lack of relevance to other types of AF such as paroxysmal AF.

I suggest a fair and honest account on these challenges and their implications on the robustness and clinical relevance of the model. This makes your results more reliable and easier to interpret.

4.Critically compare the model to the more recent literature of large-animal AFib research?

Although major historical models are mentioned, the manuscript does not provide an in-depth comparison with the more recent and emerging, including transgenic or optogenetic AFib models.

Would like to see more of comparative analysis between your model and current large animal models and incorporate knowledge of modern techniques. This should help place your work more effectively in the changing AFib research environment.

Minor Points for Revision

•Grammar and Style: Clarify sentences and eliminate slangy language (e.g., "pigs presumably feel better").

•Figures: 1) Other than captions figure quality and resolution is needed (e.g., Figure 3 needs statistics and clearly labeled axes).

•Data Availability: raw ECG and summary data should be made available to the research community in an appropriate format.

Reviewer #2: This paper compares previous AFib models with swine and focusing on key methodological differences as well as outcomes. The manuscript is organized nicely. Following comment would strengthen manuscript.

Comment 1: In the result section

•It is not clear in terms of clarity and conciseness, e.g. Persistent AFib was induced in ⅚ subjects within 60 days. It should be 5/6 along with %

•In addition, reorganization contents would help.

•It is unclear about cited reference “Pharmacologic intervention with beta-blockers and calcium channel blockers did not control rapid ventricular rates in one subject (Bishop & Akram, 2021; Pratt et al., 1983)”

•A separate Protocol adjustments section is highly recommended along with reason of amendment.

•A tabular comparison which includes animal model, clinical limitation and clinical relevance would add more weightage.

**Do you want your identity to be public for this peer review?** For information about this choice, including consent withdrawal, please see our Privacy Policy

Reviewer #1: No

Reviewer #2: **Yes: ** Prakash Sojitra, PhD

---

## [Author Response · Author response to Decision Letter 1]

16 Dec 2025

We thank the reviewers for their repsonses. We have a more detailed response to the reviewers attached for ease. Please let us know if you need anything. Thank you very much!

---

## [Decision Letter · Decision Letter 1]

30 Dec 2025

Yucatan Mini Swine Model of Atrial Fibrillation: Clinical Relevance

PONE-D-25-22478R1

Dear Dr. Goldman,

We’re pleased to inform you that your manuscript has been judged scientifically suitable for publication and will be formally accepted for publication once it meets all outstanding technical requirements.

Kind regards,

Kamal Sharma

Academic Editor

PLOS One

Additional Editor Comments (optional):

Reviewers' comments:

Reviewer's Responses to Questions

**Comments to the Author**



Reviewer #1: Yes

Reviewer #2: Yes

2. Has the protocol been described in sufficient detail?

To answer this question, please click the link to protocols.io in the Materials and Methods section of the manuscript (if a link has been provided) or consult the step-by-step protocol in the Supporting Information files.

Reviewer #1: Yes

Reviewer #2: Yes

3. Does the protocol describe a validated method?

Reviewer #1: Yes

Reviewer #2: Yes

4. If the manuscript contains new data, have the authors made this data fully available?

Reviewer #1: Yes

Reviewer #2: Yes

**5. Is the article presented in an intelligible fashion and written in standard English?**

Reviewer #1: Yes

Reviewer #2: Yes

Reviewer #1: The manuscript presents a well-designed and clinically relevant large-animal protocol for inducing persistent atrial fibrillation, addressing key limitations of existing AF models. The use of Yucatan miniswine, chronic atrial pacing, continuous rhythm monitoring, and functional activity tracking represents a significant methodological strength with high translational value. The protocol is described in sufficient technical detail to enable reproducibility by other laboratories, which is a major asset for a methods-focused publication. Validation of persistent AF, survival outcomes, and histological confirmation of atrial remodeling adequately support the robustness of the model. Although the sample size is limited, this is acceptable for a protocol development study and is transparently acknowledged. Overall, this work constitutes a valuable methodological contribution to the field and will be of interest to researchers developing and testing therapies for atrial fibrillation.

Reviewer #2: Revised manuscript is now edited appropriately and justifications cited in cover letter are sufficient.

**Do you want your identity to be public for this peer review?** For information about this choice, including consent withdrawal, please see our Privacy Policy

Reviewer #1: No

Reviewer #2: **Yes: ** Prakash Sojitra, PhD

---

## [Editor Report · Acceptance letter]

PONE-D-25-22478R1

PLOS One

Dear Dr. Goldman,

I'm pleased to inform you that your manuscript has been deemed suitable for publication in PLOS One. Congratulations! Your manuscript is now being handed over to our production team.

Kind regards,

on behalf of

Dr. Kamal Sharma

Academic Editor

PLOS One